# Seasonal Flight Pattern of the Kissing Bugs *Triatoma rubida* and *T. protracta* (Hemiptera: Reduviidae: Triatominae) in Southern Arizona, United States

**DOI:** 10.3390/insects13030265

**Published:** 2022-03-07

**Authors:** Justin O. Schmidt, Mary L. Miller, Stephen A. Klotz

**Affiliations:** 1Southwestern Biological Institute, 1961 W. Brichta Dr., Tucson, AZ 85745, USA; 2Department of Medicine, University of Arizona, 1501 N. Campbell Ave., Tucson, AZ 85724, USA; mary.gelsinger@gmail.com; 3Division of Infectious Diseases, Department of Medicine, University of Arizona, 1501 N. Campbell Ave., Tucson, AZ 85724, USA; sklotz@deptofmed.arizona.edu

**Keywords:** Chagas disease, dispersal, Sonoran Desert, moon light, allergy, *Trypanosoma cruzi*

## Abstract

**Simple Summary:**

Kissing bugs are bloodsucking insects that can transmit the dangerous and potentially lethal Chagas disease and also cause allergic reactions. They are most commonly encountered in the southwest desert (Tucson, AZ area) during the months of May through July. We wished to determine the weather conditions that were most favorable for kissing bug flight activity and, thereby, the times when people should be most careful to prevent them from entering homes and biting people. The weather factors that were most important for predicting high activity of *Triatoma rubida* were evening warm temperatures and low wind speeds. Humidity and moonlight were not important. This information is useful for inhabitants to know when to take the greatest precautions to exclude these insects from entering residences and placing them at risk.

**Abstract:**

The two most common kissing bugs, *Triatoma rubida* and *T. protracta*, in the Sonoran Desert around Tucson, Arizona are hematophagous vectors of Chagas disease and can induce potentially life-threatening allergic reactions. They were surveyed during their summer dispersal flight period to determine which environmental factors are correlated with flight activity. The two most important factors governing flights of *T.*
*rubida* were temperatures in the range of 26–35 °C and wind speeds below 14 km/h (9 miles/h). Flights were reduced below or above those temperatures, or when wind speeds exceeding 14km/h. Relative humidity and presence or absence of moonshine appeared unimportant. During their dispersal flight periods of May through July and, especially, between the peak of the flight season, 20 June to 5 July, biologists seeking to collect bugs and homeowners wishing to exclude these biting bugs from entering their homes should be most attentive during evenings of average temperature and low wind speed.

## 1. Introduction

Kissing bugs (Reduviidae: Triatominae) readily feed upon humans, their domestic animals, and a variety of wildlife including some invertebrates [1,2,3,4,5]. Their bites are not venomous *per se*, but they often induce intense local reactions and itching and sometimes dangerous, potentially fatal, allergic reactions [2,6,7,8,9]. The bugs also transmit Chagas disease which kills approximately 12,000 people per year and infects 6 million others, many of whom are severely debilitated [10]. In addition to causing human misery, Chagas disease takes a heavy toll on domestic dogs [5,11]. An estimated 70 million individuals are currently at risk of infection with Chagas disease globally [12]. After a human is infected, lifelong infection can occur if undiagnosed and untreated. Chronic Chagas disease can cause nonischemic dilated cardiomyopathy in up to a third of those infected in the US and Latin America [3,13].

In southern Arizona, USA, the most common kissing bug species are *Triatoma rubida, T. protracta* and *T. recurva*. In Tucson, Arizona, kissing bugs have a dispersal flight period mainly from mid-May through mid-July with occasional individuals flying before or after this period. They become a medical problem mainly during the hot seasonal time just before and after the beginning of the summer rains [1,8]. During this time, they fly to light sources, including lights around and inside homes, whereupon they land on the surface and often remain in the vicinity. When the sun rises, they seek shelter in dark areas and tend to crawl under doors and through gaps around windows and enter the home [8]. Kissing bugs are poor flyers and, when dispersing, are light and unfed [8,14]. We have never observed well-fed individuals fly (personal observations). Consequently, when the bugs enter homes, they are hungry and readily bite the people and pets therein.

Researchers investigating kissing bugs are often frustrated by their poor ability to predict which evenings and conditions will be best for collecting bugs and the converse of when bugs are unlikely to fly and be collected. We conducted this investigation to attempt to determine how the various factors of temperature, humidity, wind speed and moonlight might serve as cues for the bugs in the Sonoran Desert region to fly that evening. The goal was to help predict which days would be most productive surveying or collecting bugs for research or control.

## 2. Methods and Materials

### 2.1. Study Location (32°15′55.45″ N; 111° 05′05.04″ W; Elev. 800 m)

The study was conducted in the foothills of the east side of the Tucson Mountains in Pima County, Arizona. This location and the surrounding area stretching at least 10 km has not been grazed by livestock for 90 years. The immediate surroundings are nearly pristine, undisturbed habitat surrounded by county land preserves and with only five human dwellings within 500 m. Vegetation is classified as Arizona Upland Sonoran Desert [15,16] in which dominant large plant species are saguaro cacti (*Carnegiea gigantea*) (for information on this and other species, see: https://www.inaturalist.org/taxa/ accessed on 12 February 2022), little leaf paloverde (*Parkinsonia microphylla*), mesquite (*Prosopis velutina*), creosote bush (*Larrea tridentata*) and jojoba (*Simmondsia chinensis*). A wide variety of species of cacti, shrubs and bursage (*Ambrosia deltoidea*) constitute the middle vegetative zone, and numerous small ephemeral plants are present during and after periods of rainfall. Abundant small mammals, especially packrats (*Neotoma albigula* (Hartley, 1894)) and deer mice (*Peromyscus maniculatus* (Wagner, 1845)) are present at the location, which also hosts a rich diversity of large and small mammals, birds, lizards, snakes and amphibians.

### 2.2. Survey Methodology

The location is an inhabited house with large, screened sliding doors facing northwest and southeast, a large window facing northeast, another large window facing southeast, and on the south side, a 40 W fluorescent blacklight elevated above a large white sheet and facing east and west. The inside of the house was well lit with compact fluorescent lights and no blinds were drawn. Every night from 6 May 2020 through 17 July 2020, the two doors, the two large windows, and the blacklight were surveyed every 15 min for 1.5 h and all kissing bugs were captured, and their sex recorded. Environmental conditions measured consisted of the temperature, relative humidity and wind speed at 21:00. Additionally, the presence or absence of moonlight was recorded for the survey period.

### 2.3. Statistics

A generalized linear model (GLM) was used to assess the effects of covariates on the number of kissing bugs captured. A negative binomial distribution was used to account for overdispersion of the count data (number of kissing bugs recorded per night). The following explanatory variables were used in the GLM.

Wind speed—average wind speed in miles per hour (mph) over a three-hour interval (20:00–22:00);Moon—moon was visible (yes/no) the night of data collection;Relative humidity—humidity (%) measured at 21:00;Celsius—outdoor temperature measured at 21:00;Sex—sex of bug (male or female).

The estimate (log-count), 95% confidence interval and p-value for each covariate were calculated. The model was fitted using PROC GENMOD (SAS, Cary, NC 27513-2414, USA) in SAS version 9.4.

## 3. Results

We observed 340 bugs over the 70 nights in Tucson, Arizona, averaging 4.9 bugs per night. *Triatoma rubida* was more prevalent by a factor of 5.3 (*n* = 286 bugs, average = 4.1 bugs per night) than *T. protracta* (*n* = 54 bugs, average = 0.8 bugs per night). The ratio of male to female bugs was 1.3 to 1 overall with male to female ratios 1.2 to 1 and 1.8 to 1 for *T. rubida* and *T. protracta*, respectively. Over the 70-day interval, the average temperature was 31 °C, the relative humidity was 16%, and wind speed was 12.6 km per hour. The moon was visible during the survey time 36% of the nights. Figure 1 is a graphical plot of bug count by sex, temperature and moon visibility.

The only factor that significantly increased the number of *T. protracta* was moon visibility (log-count = 0.85, 95% CI: 0.22, 1.48, *p*-value = 0.01). While not statistically significant, an increase in temperature contributed to an increase in the expected log-count of male *T. protracta* while wind speed and relative humidity had a negative impact on their capture number (Table 1). In the case of *T. rubida*, all factors except sex were statistically significant. Increases in temperature, relative humidity, or wind speed negatively affected the quantity of this species, while light from the moon increased the expected log-count (Table 1) and 95% confidence intervals (Figure 2).

## 4. Discussion and Conclusions

In the Sonoran Desert around Tucson Arizona, the two most common species of kissing bugs, *Triatoma rubida* and *T. protracta*, flew to lights mainly from early May until mid-to-late July with a peak flight period from about 20 June to 5 July 2020, during which time about half of the bugs flew. Infrequently, bugs have been recorded flying as early as March and as late as late August, but these are extreme outliers. At higher elevations, such as Bisbee, Arizona (1600 m) which is twice as high as Tucson, the flight season is delayed by about a month later in the year.

During the flight season, factors that might influence the number of individuals flying were temperature, humidity, wind speed and the presence of moonlight. Flight activity decreased during hotter evening temperatures for *T. rubida* but not for *T. protracta*. Flights of both species decreased in evenings with higher wind speeds and humidities. The presence of moonlight increased the probability of both species flying, a factor that surprised us given the widespread view among entomologists that in the presence of moonlight, activities of many potential prey animals decrease because they could be seen better by their predators. Overall, warm evenings about the average temperature for the time of year and low wind speed and humidity tended to be the times of high flight activity. A limitation of this study was that only one year was surveyed. Consequently, the higher flight activity during moonlit nights was likely an artifact caused by the coincidence of the moon being present from 25 June through 5 July during the peak of flight activity in which two thirds of the bugs flew when the moon happened to be present. We suspect that other years when the moon was not present during the flight time from 20 June to 5 July that the flight pattern would be similar, if not greater than, what we observed.

Our results agree with those of other investigators who found that temperature is an important factor in the flights of *Triatoma rubida* or *T. protracta* in the southwestern parts of the USA. Sjogren and Ryckman [17] investigated *T. protracta* in an area near Redlands, California. That area was of comparable elevation to ours in Tucson, Arizona, but the temperatures in their evenings were, on average, about 6° cooler [17]. The bugs flew at times when the temperatures in their area were high within their temperature range, whereas in our area, we observed a reduction in flights during the evenings of higher temperatures. These differences likely relate to the physiology of flight requiring temperatures generally above 20 °C to become airborne and that above 30 °C, temperatures are detrimental due to excess water loss, especially in the low humidity conditions in Tucson. Ekkens [14] reported that *T. rubida* in a Tucson, Arizona area, about the same elevation as our study area, had optimal flights at temperatures of 26–35 °C, and that below 20 °C, flights ceased. His results are consistent with ours in which flights mainly ceased below 20 °C and above 35 °C.

Wood [18] wrote: “The humidity then dropped clear out of sight and with it *Triatoma* [*rubida*] disappeared until our summer rains became normal…” and suggested that high humidity was an important factor [18]. Sjogren and Ryckman reported that in their area (in which relative humidities were about four-fold higher than in our area), that relative humidity appeared not to be an important factor. Ekkens came to a similar conclusion. In our study, humidity higher than 35% occurred on only two evenings, but within our low humidity range, we observed higher humidities correlated with reduced flight activity in *T. rubida*. We feel that the report of Wood is both anecdotal and incorrect, and that humidity per se is a minor factor that may be correlated with other factors such as temperature, wind and rain that do affect flight behavior [17].

Wind speed appears to be an important factor in our investigation into the abiotic factors that affect kissing bug flight behavior. Both Sjogren and Ryckman, and Ekkens reported similar findings that bug flight decreased during evenings with high winds. These findings are consistent with the observation that kissing bugs are poor flyers in general.

Moonlight seemed irrelevant in the study by Sjogren and Ryckman [17]. Their finding supports our inclination to suspect that our results vis-à-vis moonlight are an artifact of timing of the season and the moon. To resolve this question of the importance of moonlight in the flight of kissing bugs would necessitate studies over many years to tease out moonlight from other temporal and abiotic factors, something we were unable to do.

Finally, we speculate that the flights of kissing bugs in the southwestern USA are primarily for dispersal and that their low body weights are not mainly a result of starvation, but rather of the necessity to be lightweight in order to be able to fly effectively.

## Figures and Tables

**Figure 1 insects-13-00265-f001:**
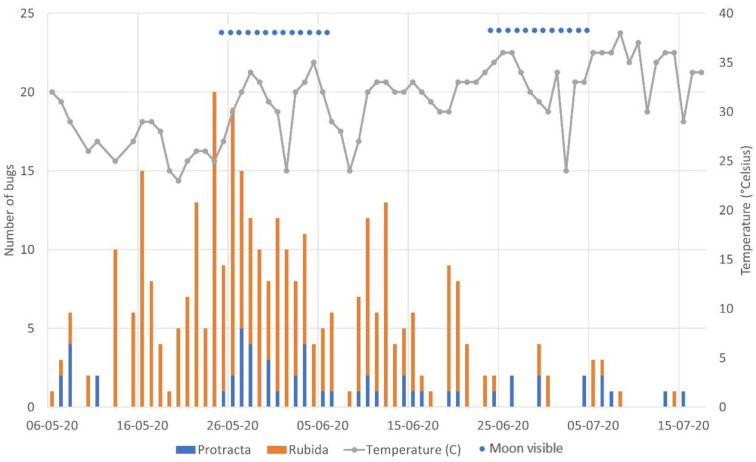
Number of bugs captured nightly from 6 May 2020 through 17 July 2020 along with nightly temperature and moon visibility.

**Figure 2 insects-13-00265-f002:**
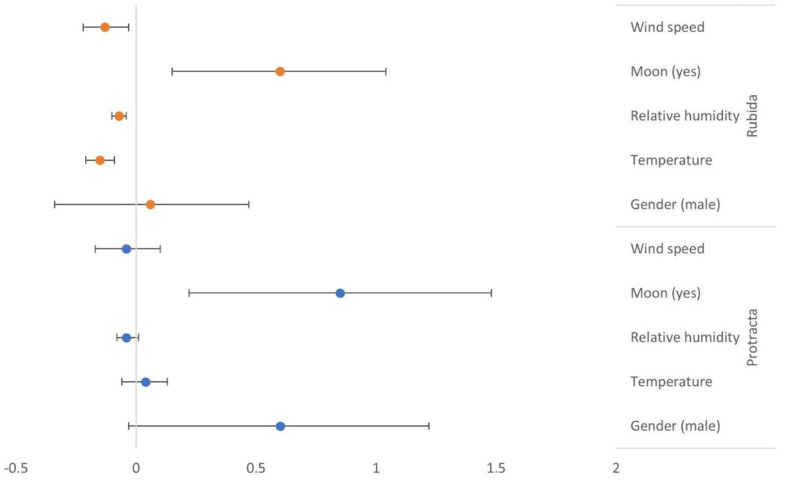
A forest plot of the log-count of kissing bugs and 95% confidence intervals for explanatory variables of wind speed, moon visibility, relative humidity (%), outside temperature and gender of bug.

**Table 1 insects-13-00265-t001:** Results of negative binomial regression for kissing bug flight count from 6 May 2020 to 17 July 2020.

	Estimate (Log-Count)	95% Confidence Interval	*p*-Value
*T. protracta*			
Sex (male)	0.60	−0.03, 1.22	0.06
Temperature	0.04	−0.06, 0.13	0.46
Relative humidity	−0.04	−0.08, 0.01	0.08
Moon (yes)	0.85	0.22, 1.48	0.01
Wind speed	−0.04	−0.17, 0.10	0.58
*T. rubida*			
Sex (male)	0.06	−0.34, 0.47	0.76
Temperature	−0.15	−0.21, −0.09	<0001
Relative humidity	−0.07	−0.10, −0.04	<0001
Moon (yes)	0.60	0.15, 1.04	0.01
Wind speed	−0.13	−0.22, −0.03	0.01

Covariates: Sex (male vs. female), temperature (degrees Celsius), relative humidity (% humidity), moon (moon visibility—yes/no), wind speed (average wind speed over a three-hour time span).

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
