# Peer review of "Seasonal Flight Pattern of the Kissing Bugs Triatoma rubida and T. protracta (Hemiptera: Reduviidae: Triatominae) in Southern Arizona, United States"

_insects, 2022, doi:10.3390/insects13030265_

Round 1

Reviewer 1 Report

This short article describes migration of two local species of kissing bugs over one season, approximately 10 weeks, by measuring how many bugs were captured at one house for 1.5 hours every evening during the migration season. By comparing the number of kissing bugs captured and the conditions (wind speed, moon visibility, humidity, temperature) and sex of the bug allowed the researchers to correlate migration with conditions and sex. Results will be useful for further research and in public health communication.

As this is an article describing observations of just one season of kissing bug migration, in one locality, I recommend it be published as a note or short communication. Also, limitations of the study should be included in the discussion.

The language that expresses causation, “inducers of fight activity”, “suppressed flights” and “depressed flights” should be modified to indicate correlation, not causation, since they don’t present controlled experiments. The literature indicates kissing bug flights are often induced by starvation or the need to mate. As they are reporting observations, the article should be written indicating more flights were observed at particular temperatures and wind speeds, not that these factors “induced”, “suppressed” or “depressed”.

The simple summary and abstract need to be rewritten separating the results by species, which gave quite different results. In the discussion, they should make only the conclusions that show statistical significance and specify to which species this refers, e.g., ln. 136-137; 142-144.

Author Response

We thank you for your excellent suggestions for improving the manuscript.

We concur with your recommendation to change the category from Article to Short Communication.

We also included in the discussion information about the limitations of this study.

The reviewer correctly pointed out that our findings are correlations and not causations; thus, we have changed the wording to clarify that they are, indeed, correlations.

The simple summary and abstract now reflect the differences observed between the two species and the discussion emphasizes conclusions that are statistically significant.

Detailed responses please see the attachment.

Reviewer 2 Report

I made few considerations that are in the attached pdf. After the modifications, the manuscript will be recommended for publication on my part.

Congratulations on the design. I believe it will serve as a model for studies in South America.

Author Response

We thank the reviewer for thorough and helpful suggestions and editing recommendations.  We have incorporated the changes suggested and are most appreciative for improving the manuscript.

Detailed responses please see the attachment.
